# CSF Heavy Neurofilament May Discriminate and Predict Motor Neuron Diseases with Upper Motor Neuron Involvement

**DOI:** 10.3390/biomedicines9111623

**Published:** 2021-11-05

**Authors:** Cecilia Simonini, Elisabetta Zucchi, Roberta Bedin, Ilaria Martinelli, Giulia Gianferrari, Nicola Fini, Gianni Sorarù, Rocco Liguori, Veria Vacchiano, Jessica Mandrioli

**Affiliations:** 1Department of Biomedical, Metabolic and Neural Sciences, University of Modena and Reggio Emilia, 41125 Modena, Italy; ceciliasimonini24@gmail.com (C.S.); bedin.roberta@aou.mo.it (R.B.); gianferrari.giulia@gmail.com (G.G.); 2Neurology Unit, Azienda Ospedaliero Universitaria di Modena, 41126 Modena, Italy; martinelli.ilaria88@gmail.com (I.M.); fini.nicola@aou.mo.it (N.F.); 3Clinical and Experimental PhD Program, University of Modena and Reggio Emilia, 41125 Modena, Italy; 4Neuromuscular Center, Department of Neurosciences, University of Padova, 35121 Padova, Italy; gianni.soraru@unipd.it; 5Clinica Neurologica, Azienda Ospedaliera di Padova, 35128 Padova, Italy; 6IRCCS Istituto delle Scienze Neurologiche, Ospedale Bellaria, 40139 Bologna, Italy; rocco.liguori@unibo.it (R.L.); veriavacchiano@gmail.com (V.V.); 7Department of Biomedical and Neuromotor Sciences, University of Bologna, 40127 Bologna, Italy

**Keywords:** motor neuron disease, neurofilaments, upper motor neuron, degeneration

## Abstract

*Objective:* To assess whether phosphorylated neurofilament heavy chain (pNfH) can discriminate different upper motor neuron (UMN) syndromes, namely, ALS, UMN-predominant ALS, primary lateral sclerosis (PLS) and hereditary spastic paraparesis (hSP) and to test the prognostic value of pNfH in UMN diseases. *Methods:* CSF and serum pNfH were measured in 143 patients presenting with signs of UMN and later diagnosed with classic/bulbar ALS, UMNp-ALS, hSP, and PLS. Between-group comparisons were drawn by ANOVA and receiver operating characteristic (ROC) analysis was performed. The prognostic value of pNfH was tested by the Cox regression model. *Results:* ALS and UMNp-ALS patients had higher CSF pNfH compared to PLS and hSP (*p* < 0.001). ROC analysis showed that CSF pNfH could differentiate ALS, UMNp-ALS included, from PLS and hSP (AUC = 0.75 and 0.95, respectively), while serum did not perform as well. In multivariable survival analysis among the totality of UMN patients and classic/bulbar ALS, CSF pNfH independently predicted survival. Among UMNp-ALS patients, only the progression rate (HR4.71, *p* = 0.01) and presence of multifocal fasciculations (HR 15.69, *p* = 0.02) were independent prognostic factors. *Conclusions:* CSF pNfH is significantly higher in classic and UMNp-ALS compared to UMN diseases with a better prognosis such as PLS and hSP. Its prognostic role is confirmed in classic and bulbar ALS, but not among UMNp, where clinical signs remained the only independent prognostic factors.

## 1. Introduction

Syndromes presenting with upper motor neuron (UMN) signs include diseases with very variable prognosis, ranging from “pure” UMN diseases (primary lateral sclerosis and hereditary spastic paraplegia) to UMN-predominant presentation of amyotrophic lateral sclerosis (ALS) [1]. 

Recently, the diagnostic criteria for primary lateral sclerosis (PLS) were revised to facilitate an earlier diagnosis of the disease. Although the observation interval has been shortened, at least two years are still needed to define probable PLS [2]. PLS is characterized by a distinctly longer survival than ALS, while UMN-predominant ALS (UMNp-ALS), a phenotype of ALS with a predominance of pyramidal over lower motor neuron (LMN) signs, has an intermediate prognosis between classic ALS and PLS [2,3]. The difficulty in discriminating between these two clinical entities contributes to a significant delay in the diagnosis of PLS [2]. Several lines of evidence from neuroimaging studies are currently stressing how PLS is a demarcated disease entity from ALS, thus advocating separate therapeutic opportunities [4], while UMNp-ALS is a well-defined phenotype of ALS, sharing the same pathomechanisms of the other clinical presentations. On the other hand, diagnosis of hereditary spastic paraplegia (hSP) is based on slowly progressive UMN symptoms traditionally confined to lower limbs, family history and genetics, with more than 70 genetic subtypes described [5]. Furthermore, hSP can be distinguished from PLS due to earlier onset and bulbar sparing, but notable exceptions exist [6]. 

Among promising ALS biomarkers, neurofilament light chain (NfL) and phosphorylated neurofilament heavy chain (pNfH) have proved to discriminate well between cases and healthy controls [7]. Neurofilaments are major axonal cytoskeleton proteins that are essential for the structural stability of myelinated axons and for reaching the optimal propagation speed of electrical impulses. These proteins are released from damaged or diseased axons both in cerebrospinal fluid (CSF) and blood and have been proposed as diagnostic or prognostic biomarkers for ALS and other neurodegenerative conditions [8,9]. 

In the present study, we measured pNfH levels in CSF and serum of patients presenting with UMN signs, who were later diagnosed within a definite clinical group (classic or bulbar ALS, UMNp-ALS, PLS, hSP) with subsequent diagnostic confirmation after a long follow-up period (median: 85 months). This study was designed to investigate whether pNfH can be used as a complementary tool for differential diagnosis in patients presenting with UMN syndromes and to help reduce diagnostic delay in UMN syndromes. 

## 2. Materials and Methods

### 2.1. Study Design, Standard Protocol Approval, and Recruitment

This retrospective study included patients who underwent lumbar puncture as part of their diagnostic work-up between 1 January 2007, and 1 January 2019. The Ethical Committee of Area Vasta Emilia Nord approved the study, which was performed according to the declaration of Helsinki. 

### 2.2. Participants 

Among a total of 281 patients diagnosed with MND with UMN involvement during the study period, 143 patients underwent lumbar puncture during the diagnostic process and still had at least 0.5 mL of CSF and serum available when the present study was designed. All patients presented with signs of UMN syndrome as defined by a combination of hyperreflexia, spasticity, clonus, and pathological reflexes at the time of the diagnostic work-up. Patients were followed up with a multidisciplinary approach for a median of 85 months and finally diagnosed with definite or probable ALS (El Escorial revisited criteria)—classic or bulbar phenotype—(95 patients) and UMNp ALS (20 patients). The phenotypic distinction among flail or respiratory forms was also taken into account in the analysis according to established criteria [3]. A definite diagnosis of PLS (15 patients) or hSP (13 patients) was made according to recently published criteria [2] for the former and based on confirmed mutations, family history, or long-term clinical follow-up for the latter [5]. Clinical assessment was performed by specialized MND clinicians blinded to neurofilament measurements.

As part of our systematized neurological assessment, for this study we considered several clinical signs of MN dysfunction that were detected at sampling time: fasciculations frequency and distribution, presence of Babinski and Hoffman signs as well as Achilles clonus (absent, monolateral, bilateral), exaggerated masseter reflex, presence of frontal release signs (palmomental, snout and glabellar reflex), pseudobulbar affect (measured by CNS-lability score > 13) [10] and presence of cramps. Spasticity at the time of sampling and at the last observation was assessed using the Ashworth scale in the most affected limb. The burden of UMN signs was considered by retrospective calculation of the recently validated Penn UMN score [11]. Cognitive and behavioral alterations in the frontotemporal spectrum disorder were recorded according to revised Strong’s criteria [12]. Fasciculations frequency and distribution were classified by modifying an established score employed for muscle ultrasonography [13,14]: 0 was assigned for no observable or sporadic fasciculations after one-minute observation, 1 for single fasciculation observed in one muscle, 2 for focal continuous fasciculations (i.e., fasciculations localized in one muscular district), 3 for multifocal continuous fasciculations. 

Assessment of clinical progression was performed using the ALS Functional Rating Scale Revised (ALSFRS-R) (total score and 4 subscores from its four main domains) [15] through time, though this latter scale has not been validated for hSP and PLS. Progression rate was assessed at two time points, at baseline according to the original work by Kimura [16], and at last observation (PRL), taking into account the difference between the last observable ALSFRS-r score and 48, over the time period from last visit to the referred onset of disease [17]. Time to generalization was defined as the time in months between the spread of motor weakness from one district (considering bulbar, cervical, thoracic, and lumbar segments) to another. 

Corticospinal involvement at brain MRI was assessed in 118 patients as the presence of corticospinal tract hyperintensity in FLAIR or diffusion tensor images; motor evoked potentials were assessed in 114 patients and the presence of prolongation of central conduction time was considered as a sign of corticospinal involvement. 

We did not include healthy controls (HC) since diagnostic difficulties are usually present when facing patients with variably distributed UMN signs and not with HC, and since pNfH have a known capacity to discriminate between MND and HC [18]. 

### 2.3. Sample Collection and Analysis

CSF and serum samples were collected at the same time and underwent separate processing according to standard international procedures [19]. 

CSF pNfH levels were determined using a CE-marked ELISA test with polyclonal capture and monoclonal detection antibodies (Euroimmun, Lübeck, Germany), as previously reported [8]; serum pNfH levels were determined using a high-sensitivity CE-IVD (CE-In Vitro Diagnostic) marked ELISA test (Euroimmun), as previously reported [8]. Each sample was measured in duplicate and 2 quality controls with known low and high concentrations, respectively, were measured in each plate.

### 2.4. Statistics

Density analysis of pNfH concentrations in CSF and serum revealed an extremely left-skewed, non-normal distribution (skewness/kurtosis tests for normality: *p* < 0.001 both for CSF and serum pNfH). After cube root transformation, data appeared to be normally distributed, thus cube root-transformed pNfH (crt-pNfH) concentrations were used for graphical representation of the data and analysis of variance models.

crt-pNfH levels were compared among different diagnostic groups (ALS, UMNp ALS, PLS, and hSP) and ALS phenotypes (classic, bulbar, UMNp, flail, respiratory) using ANOVA, with the Tukey–Kramer test for pairwise comparisons between groups. The influence of age, disease duration and sex on crt-pNfH within each group and overall was tested by ANCOVA. Correlations between crt-pNfH and quantitative variables (CSF storage time, disease duration, disability scales scores, ΔFS, Penn UMN score, fasciculation grading) were assessed using Pearson’s correlation coefficient and linear regression models. Receiver operating characteristic (ROC) curves were used for a sensitivity analysis of pNfH cutoff values in CSF and serum for ALS versus other UMN diseases (PLS and hSP), for UMNp ALS versus PLS, for PLS versus hSP, and UMNp ALS versus hSP. Overall sensitivity and specificity were evaluated by areas under the curve (AUC). 

Survival (months from onset to death/tracheostomy, and with live patients censored) was assessed first with univariate analysis using a Cox proportional hazards regression model, Kaplan–Meier curves, and log-rank test. The following variables were examined: crt-pNfH levels in serum and CSF, MND subtypes, age at onset, sex, diagnostic delay, time from onset to sampling, clinical MN signs, ALSFRS-R total score and subscores (bulbar, upper limb and lower limb motricity, and respiratory), and Ashworth scale score. A multivariable analysis of survival with a Cox proportional hazards regression model was then performed with the abovementioned variables, which reached a significance level (*p* < 0.1) using the stepwise backward method. HR were calculated for each variable with 95% CI. All statistical analysis was performed using the software STATA version 14.0 (StataCorp. 2015. College Station, TX, USA: StataCorp LP, USA).

## 3. Results

### 3.1. Patients Characteristics

Table 1 shows the clinical features of patients at sampling and at diagnosis, considering UMN-p ALS (*n* = 20) apart from the other phenotypes of ALS, who were classified as following: 46 classic ALS, 20 bulbar ALS, 10 flail (including both flail arm and flail leg) and 3 respiratory ALS. 

Among the 13 hSP patients, 6 had a genetic confirmation. The median follow-up from sampling to the last visit (or date of death/tracheotomy) and the progression rate reflect the different prognoses of each group. There were obvious, different functional disability scores across the evaluated scales among groups. Clinical and instrumental signs of motor neuron impairment across different MND were considered with reference to the neurological examination at the time of sampling and are summarized in Table 2. 

### 3.2. CSF and Serum pNfH Concentrations in Different MND

Median concentrations of pNfH in CSF and serum across the four diagnostic group are shown in Table 3. 

UMNp ALS patients and ALS patients with other clinical phenotypes (classic, bulbar, flail and respiratory) had similar levels of crt-pNfH in CSF and serum (*p* = 0.827 and *p* = 0.746 at one-way ANOVA, respectively). ALS patients had higher levels of crt-pNfH compared to hSP (*p* < 0.001 for both CSF and serum) and PLS, with the latter only reaching significance in CSF (*p* < 0.001 and *p* = 0.07 for CSF and serum, respectively). Similar results were obtained when analyzing UMNp ALS with PLS and hSP (*p* > 0.001 for pairwise comparisons between both groups in CSF analysis and only hSP in serum, while *p* = 0.06 for the comparison with PLS in serum). PLS patients had higher levels of crt-pNfH in CSF and serum compared to hSP (*p* = 0.030 and *p* = 0.013 in CSF and serum). The distribution of crt-pNfH in CSF and serum across ALS, UMNp ALS, PLS, and hSP is summarized in Figure 1. 

In serum pNfH, significant differences were observed only between ALS, UMNp and hSP while concentrations of CSF pNfH were significantly different for ALS and PLS, ALS and hSP, UMNp and PLS, and UMNp and hSP. 

The ANCOVA excluded significant effects of sex or age at sampling on crt-pNfH concentrations, whereas disease duration affected crt-pNfH (F[1, 136] = 19.99, *p* < 0.001 for CSF; F[1, 136] = 8.05, *p* = 0.005 for serum). The analysis was next repeated within diagnostic groups, confirming a significant effect of MND groups after correction for age, disease duration, and sex on CSF (F[4, 136] = 27.26, *p* < 0.001) and serum (F[4, 136] = 11.10, *p* = 0.011) concentrations of crt-pNfH.

The AUC of ROC curves for discriminating UMN syndromes are reported in Table 4.

Overall, ALS with all phenotypes—UMNp included—could be differentiated from PLS by CSF pNfH with an AUC of 0.75 (CI: 0.61–0.89) and by serum pNfH with an AUC of 0.66 (CI: 0.52–0.80). The discriminatory performance of pNfH between ALS (including UMNp ALS) and hSP was more evident both in CSF (AUC = 0.95, CI: 0.90–0.99) and serum (AUC = 0.86, CI: 0.77–0.95). Figure 2 shows the ROC curves of CSF pNfH for patients with ALS and hSP, UMNp-ALS and hSP.

### 3.3. Correlations of CSF and Serum pNfH Concentrations with Clinical Data and Disability Scores

Patients with more frequent and widespread fasciculations had higher crt-pNfH, with more evident differences in CSF (*p* = 0.007 for single fasciculations; *p* < 0.001 for single multifocal fasciculations; *p* = 0.005 for multifocal and continuous fasciculations with respect to patients without fasciculations, Figure 3) [14].

These results were corroborated by logistic regression models where crt-pNfH in CSF and serum were input as the dependent variables and fasciculation grading as the independent variable, obtaining a coefficient of 0.09 (CI: 0.06–0.13, R-square: 0.17, *p* < 0.0001) and 0.24 (CI: 0.05–0.44, R-square: 0.044, *p* = 0.02), respectively. On the other hand, crt-pNfH did not correlate with any sign of UMN dysfunction, Penn UMN score included.

The Spearman’s correlation coefficient for serum and CSF concentrations of crt-pNfH was 0.67 (*p* < 0.01). There was a correlation between concentrations of crt-pNfH in CSF and serum and diagnostic delay (ρ = −0.42, *p* < 0.001, and ρ = −0.28, *p* < 0.001, respectively), progression rate at sampling (ρ = 0.596, *p* < 0.001, and ρ = 0.42, *p* < 0.001, respectively), progression rate at last observation (ρ = 0.396, *p* < 0.001, and ρ = 0.227, *p* = 0.015, respectively) and total ALSFRS-R score at sampling (ρ = −0.27, *p* = 0.002, and ρ = −0.25, *p* = 0.007, respectively).

Serum and CSF concentrations of sqrt-pNfH did not correlate with time from sampling to pNfH measurement nor with age at sampling.

### 3.4. Serum and CSF pNfH and Survival

As expected, survival was extremely different among MND subgroups: 50% of patients with ALS died after 37.13 months, 50% of patients with UMNp-ALS died after 67.53, whereas more than 50% of patients with PLS were alive at the end of the observation period. As far as ALS is concerned, 10% of patients died after 13.07 months, 10% of UMNp-ALS patients died after 30.06 months, whereas only 10% of PLS died after 84.13 months. All hSP patients were alive at last observation.

Table 5 shows the univariate analysis of survival through a Cox regression model in the totality of UMN syndrome patients.

When multivariate regression analysis was performed on the variables that were significant at univariate analysis, the following remained in the model: CSF concentration of pNfH (HR 13.93, *p* = 0.001; 95% CI 2.93–66.18), time to generalization (HR 0.97, *p* = 0.006; 95% CI 0.96–0.99), presence of dementia (HR 4.03, *p* < 0.001; 95% CI 1.89–8.59), ALSFRS-R score at sampling (HR 1.10, *p* = 0.001; 95% CI 1.04–1.17), progression rate (HR 2.63, *p* < 0.001; 95% CI 1.77–3.91) and diagnostic group (HR 0.14, *p* < 0.001; 95% CI 0.06–0.33).

Then, separate survival analysis was carried out for ALS and UMNp ALS. Univariate analysis of survival through a Cox regression model among ALS patients showed the impact of the following variables: CSF concentration of pNfH (HR 26.21, *p* < 0.001; 95% CI 7.36–93.36), serum pNfH (HR 1.33, *p* = 0.004; 95% CI 1.09–1.62), time from onset to sampling (HR 0.92, *p* < 0.001; 95% CI 0.89–0.96), time to generalization (HR 0.96, *p* < 0.001; 95% CI 0.94–0.98), diagnostic delay (HR 0.93, *p* < 0.001; 95% CI 0.89–0.96), and progression rate, (HR 1.88, *p* < 0.001; 95% CI 1.49–2.37). All other variables, including age and site at onset, the presence of signs of upper or lower motor neuron, genetics, and cognitive/behavioral impairment did not affect survival. When multivariate regression analysis was performed, of the variables that were significant at univariate analysis, only CSF concentration of pNfH (HR 7.89, *p* = 0.001; 95% CI 2.27–27.52) and time to generalization (HR 0.96, *p* = 0.001; 95% CI 0.93–0.98) remained in the model.

Univariate analysis of survival through a Cox regression model among UMNp-ALS patients interestingly revealed the effect of the CSF concentration of pNfH (HR 385.08, *p* = 0.062; 95% CI 0.75–197832.8), diagnostic delay (HR 0.78, *p* = 0.036; 95% CI 0.62–0.98), progression rate (HR 2.96, *p* = 0.012; 95% CI 1.27–6.87), presence of abundant fasciculations (HR 4.94, *p* = 0.010; 95% CI 1.47–16.55), and presence of FTD (HR 14.49, *p* = 0.059; 95% CI 0.91–231.77). When multivariate regression analysis was performed, only progression rate (HR 4.71, *p* = 0.011; 95% CI 1.43–15.53) and presence of abundant fasciculations (HR 15.69, *p* = 0.018; 95% CI 1.61–153.10) preserved their role in the model. Due to the low number of deaths among PLS and hSP, survival analysis was not performed in these diagnostic groups.

## 4. Discussion

The first notable result of this study is that CSF and serum concentrations of pNfH may aid in the differentiation of UMN syndromes with a more favourable prognosis, as reported in our previous pilot study [8]. Both serum and CSF pNfH levels can discriminate ALS and UMNp from “pure UMN” with slower progression, such as hSP, as also reported by Wilke and colleagues [20] and us [8]. The distinction in pNfH levels between ALS and PLS remains less defined, though more evident at CSF analysis, in contrast with what was previously reported by Verde and colleagues [21] where pNfH was not significantly different in PLS compared to ALS. As neurofilaments are known to vary during the course of ALS, with levels rising in the pre-diagnostic phase and depending on the progression rate during the early period, we might hypothesize that the contrasting results between our and Verde’s reports might come from the different times at which samples were collected. To measure pNfH at a single time point during the diagnostic work-out may leverage patients with a fast or slow disease course and result in an average higher or lower concentration of pNfH. Importantly, in our study we considered UMNp-ALS, whose clinical presentation and differential diagnosis with PLS in the early stages might be challenging, even for the most experienced clinicians. Another recent study showed that serum pNfH differs between MND phenotypes as defined by Chiò [3], and in particular, classic and pyramidal ALS have higher levels compared to “pure upper motor neuron” forms; unfortunately, in that study ROC curve analysis was not reported for direct analysis of the discriminatory capacity [22]. This result, if confirmed by future studies, might have a significant impact on clinical practice as it would allow a faster differential diagnosis between PLS and UMNp-ALS and it would reduce the diagnostic delay for PLS. The importance of performing a lumbar puncture is of further consideration for the diagnostic process in patients presenting with UMN symptoms as well. As a further point, our results may support the notion that UMN-p ALS has similar pNfH levels compared to the other phenotypes of ALS, flail forms included, whereas PLS seems to entail a neuropathological process in which the speed and the severity of the axonal damage, as well as the dynamics of neurofilament synthesis, degradation and release in CSF and blood are different from those in the ALS spectrum.

The difference between pNfH concentrations in serum and CSF has been reported in numerous other studies [8,23,24] and may reflect the analytical capacity of the two biofluids. Although the reason is not yet well defined, it is possible that pNfH tend to aggregate in serum [23,25], reducing their diagnostic performance despite being more stable and less susceptible to degradation by proteases compared to the NfL [25]. Furthermore, an immune response to neurofilaments in the plasma of patients with ALS has been reported and this response would increase the speed of NF clearance and/or have a masking effect over the epitopes recognized by the antibodies employed in ELISA [26]. Another possibility is that pNfH ELISA are less sensitive to serum pNfH concentrations, as a comparative study between different immunoassays showed, with a lower serum-CSF correlation with respect to homebrew and commercial SiMoA [27,28]. As these new immunoassays are able to detect proteins in biological fluids even when present at low levels, and since blood is a more accessible biofluid and a less invasive matrix that can be easily collected, these techniques should be encouraged for longitudinal analysis of neurofilaments.

Our study also shed light on an interesting correlation between clinical signs of MN dysfunction and pNfH levels. In particular, patients with more frequent and widespread fasciculations had higher levels of pNfH in CSF (but not in serum). Furthermore, we confirmed a negative correlation between concentrations of pNfH in CSF and serum and diagnostic delay and a positive correlation between concentrations of pNfH in CSF and serum and progression rate at sampling [8,21]. Nfs, being the main components of the axonal cytoskeleton, are more conspicuously released by large-diameter and long neurons [18]. These results may be explained considering that for ALS the higher levels of pNfH are due to the more conspicuous neurodegenerative process involving both lower and upper motor neurons, while for PLS and hSP there is a slower degeneration of only corticospinal axons [21]. Previous studies have already proved how CSF NfL [29] and NfH [30] relate to the burden of UMN and LMN involvement, as assessed by the number of regions affected. However, the degree of affection of the two systems is extremely difficult to quantify homogenously in vivo: diffusion tensor imaging (DTI) alterations along the corticospinal tract and MEP parameters may well detect UMN disruption [31,32], but quantification of the involvement is approximate. Furthermore, currently there is no method to account for the degree of degeneration of the LMN system over the whole body, since neurophysiological techniques such as electromyography and motor unit number estimation rely on analysis of subgroups of muscles.

The significance of the relationship between neurofilament concentrations and motor neuron affection is not just a matter of pathobiology for ALS but it is also reinforced by the survival analysis. Indeed, whereas in multivariable survival analysis for ALS, pNfH sustains the model as a strong independent predictor of prognosis together with time to generalization, as also reported by Li and colleagues [33], in UMN-p ALS, the value of the pNfH declines in favor of a clinical sign such as the degree of fasciculations. As already mentioned, this might be partly explained by the lower sensitivity of ELISA when measuring pNfH in a complex matrix such as serum, and thus peripheral damage is less accurately assessed, or perhaps NF release better reflects UMN burden and peripheral markers are warranted [34].

The main drawback of this study is the small sample size, particularly for the PLS and hSP groups, which may be addressed by larger multicentric studies to overcome the rarity of these conditions. Furthermore, the retrospective design and the consequent different times at which samples were collected and stored may have impacted measurements. Longitudinal clinical, neurophysiological and biological assessment would help in better understanding Nfs correlations with UMN and LMN burden of disease. Next, we recognize that the application of neurofilaments in the clinical practice of MND clinics is still limited by several factors, first and most important, the notion that these molecules are rather aspecific markers of neuroaxonal injury, and may be elevated in other degenerative conditions or for ancillary causes such as previous trauma [35]. As already mentioned, there are still technical laboratory issues possibly impacting on pNfH quantification in ELISA assays compared to more recent and sensitive immune-assays. These analytical considerations are best assessed by international round robins, as recently published [36], in order to ensure a standardized and optimized quality for pNfH measurements across centers.

## 5. Conclusions

In conclusion, despite some statistical instability in the results due to the limitations of the sample size, our study supports the hypothesis that CSF pNfH may aid in the differential diagnosis of ALS, UMNp-ALS and PLS. This would represent a significant advance in the early diagnosis of PLS, which, despite the introduction of the most recent diagnostic criteria, still requires several years and clinical and neuropathological validation [21]. In addition, the clinical imprecision in the diagnosis, along with some uncertainty about overlap with UMNp-ALS, has become an obstacle for therapeutic development for PLS [2].

The potential role of CSF pNfH as prognostic biomarkers for MND presenting with UMN signs merits further study with a larger population and longitudinal follow-up of PLS and hSP.

Finally, the possibility of using analytical methods that are more sensitive than ELISA for serum pNfH could allow us to really evaluate the potential of this biomarker on a less invasive matrix that could also be collected longitudinally.

## Figures and Tables

**Figure 1 biomedicines-09-01623-f001:**
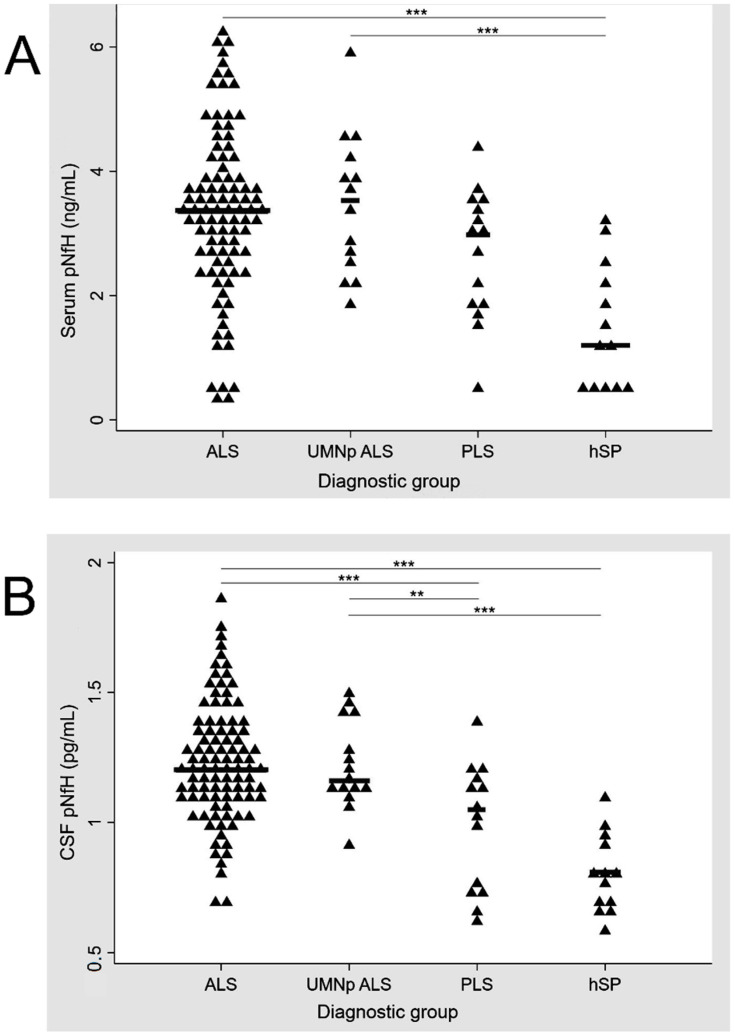
Distribution of pNfH in serum (**A**) and CSF (**B**) across ALS, UMNp ALS, PLS, and hSP. *p*-values for comparisons between diagnostic groups: *** *p* < 0.001, ** *p* = 0.05. Scatter plot of single concentrations (represented by triangles), according to the diagnostic groups. ALS, amyotrophic lateral sclerosis; CSF, cerebrospinal fluid; hSP, hereditary spastic paraparesis; pNfH, phosphorylated neurofilament heavy chain; UMNp-ALS, upper motor neuron predominant-ALS.

**Figure 2 biomedicines-09-01623-f002:**
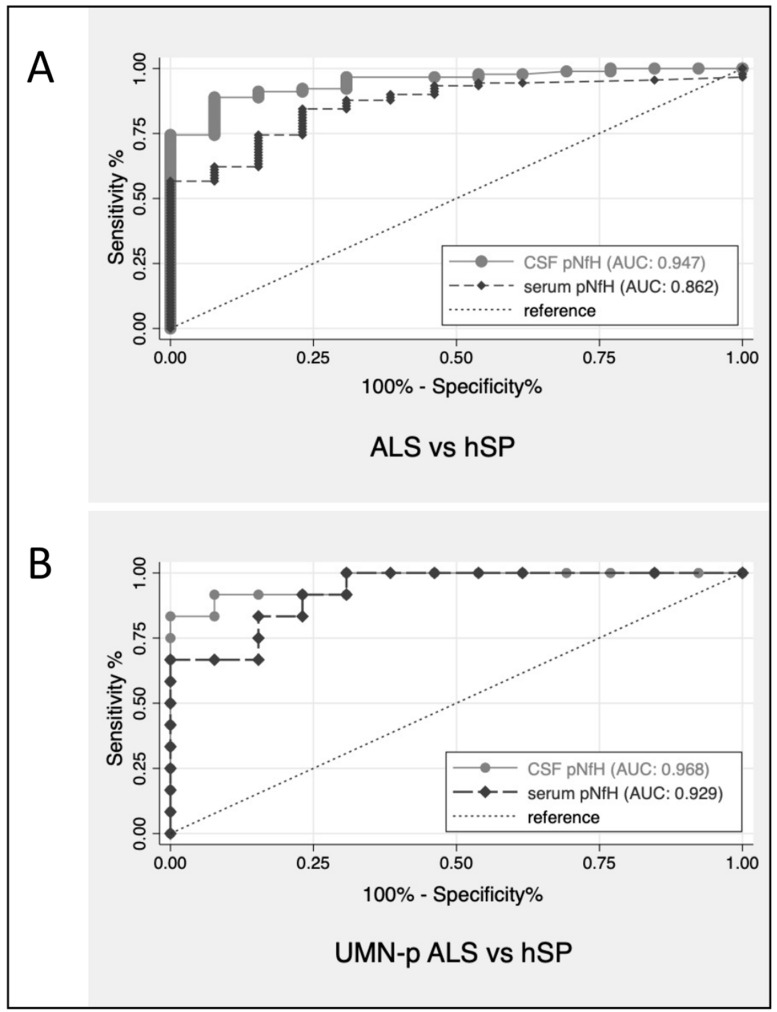
ROC curves for CSF pNfH for discrimination of patients with ALS versus hSP (**A**) and UMNp-ALS versus hSP (**B**). ALS, amyotrophic lateral sclerosis; AUC, area under the ROC curves; CSF, cerebrospinal fluid; hSP, hereditary spastic paraparesis; pNfH, phosphorylated neurofilament heavy chain; ROC, receiver operating characteristic; UMNp-ALS, upper motor neuron predominant-ALS.

**Figure 3 biomedicines-09-01623-f003:**
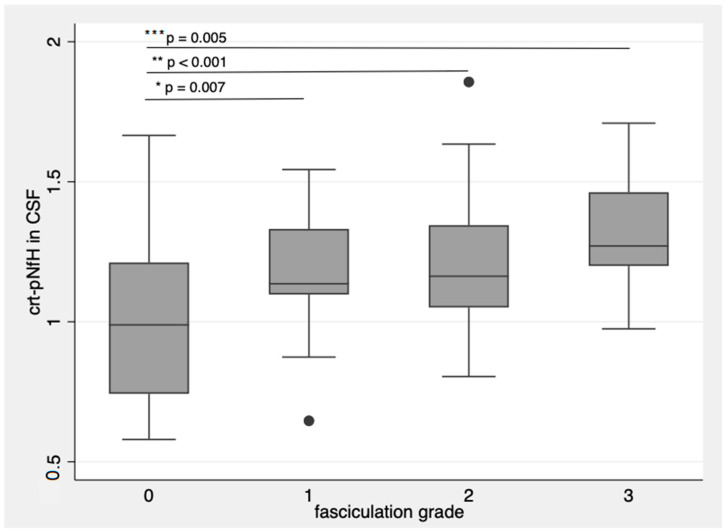
crt-pNfH in CSF concentrations raise with more frequent and diffuse fasciculations. * *p*-value = 0.007; ** *p*-value < 0.001; *** *p*-value = 0.005. Dots are outlier values of crt-pNfH in CSF according to the groups defined by fasciculation grading. The box plots show the crt-pNfH concentrations (median and interquartile values) according to the grade of fasciculations. 0: no observable or sporadic fasciculations, 1: single fasciculation observed in one muscle, 2: focal fasciculations, localized in one muscular district, 3: for multifocal continuous fasciculations.

**Table 1 biomedicines-09-01623-t001:** Subjects’ clinical features in different MND.

	ALS (*n* = 95)	UMNp ALS (*n* = 20)	PLS	hSP	*p* Value
			(*n* = 15)	(*n* = 13)	
Male sex	59 (62.11)	12 (60)	7 (46.67)	7 (53.85)	0.680
Age at sampling, years	62.2 ± 12.1	53.02 ± 12.75	63.22 ± 9.50	51.60 ± 18.84	0.010
BMI at sampling	24.64 ± 4.29	23.78 ± 3.45	24.39 ± 3.24	22.26 ± 10.48	0.565
Onset to sampling, months	12.57 ± 10.57	16.46 ± 18.14	59.29 ± 63.80	133.76 ± 131.21	<0.001
Diagnostic latency, months	14.23 ± 12.12	13.51 ± 10.88	47.61 ± 39.94	136.65 ± 130.24	<0.001
ALSFRS-R at sampling, total score	40.94 ± 5.71	39.25 ± 10.13	40.85 ± 5.43	43.90 ± 2.13	0.857
ALSFRS-R at sampling, bulbar score	10.44 ± 2.16	10.69 ± 2.68	10.69 ± 2.39	11.80 ± 0.42	0.315
ALSFRS-R at sampling, upper limbs subscore	6.33 ± 1.81	6.69 ± 2.21	6.31 ± 1.65	7.80 ± 0.63	0.099
ALSFRS-R at sampling, lower limbs subscore	5.83 ± 2.25	4.06 ± 1.84	5.08 ± 1.38	4.60 ± 1.71	0.009
ALSFRS-R at sampling, respiratory subscore	11.36 ± 1.90	11.31 ± 2.75	12.00 ± 0.00	11.90 ± 0.32	0.575
Progression rate at sampling, (points/month)	0.98 ± 1.07	0.81 ± 0.88	0.18 ± 0.10	0.09 ± 0.09	0.004
Progression rate at last observation, (points/month)	1.11 ± 1.24	0.39 ± 0.30	0.23 ± 0.16	0.16 ± 0.16	0.002
Time to generalization	13.96 ± 15.01	18.50 ± 23.24	31.81 ± 21.01	192.03 ± 186.68	<0.001

Values are expressed as means ± SD or absolute numbers (%).

**Table 2 biomedicines-09-01623-t002:** Clinical and instrumental signs of MN involvement across different MND.

	ALS (*n* = 95)	UMNp ALS (*n* = 20)	PLS (*n* = 15)	hSP (*n* = 13)	*p* Value
Pseudobulbar affect (presence)	19 (19.38)	5 (29.41)	3 (20.00)	0 (0)	0.233
Behavioural changes (presence)	15 (15.30)	0 (0)	0 (0)	0 (0)	0.040
Cognitive changes (presence)	16 (16.32)	1 (5.88)	1 (6.67)	0 (0)	0.187
Dementia (presence)	14 (14.28)	1 (5.88)	0 (0)	0 (0)	0.130
Palmomental reflex (presence)	17 (17.34)	1 (5.88)	2 (11.11)	0 (0)	0.171
Glabellar reflex (persistence)	7 (7.14)	0 (0)	3 (20.00)	0 (0)	0.173
Snout reflex (presence)	21 (21.43)	4 (23.53)	4 (26.67)	0 (0)	0.321
Masseter reflex (exaggerated)	16 (16.32)	4 (23.53)	4 (26.67)	0 (0)	0.397
Unilateral Hoffmann sign	14 (14.29)	2 (11.76)	6 (40.00)	1 (7.69)	0.003
Bilateral Hoffmann sign	17 (17.34)	9 (52.94)	5 (33.33)	3 (23.08)	
Unilateral Babinski sign	20 (20.41)	1 (5.88)	2 (11.11)	2 (15.38)	<0.001
Bilateral Babinski sign	13 (13.27)	13 (76.47)	9 (60.00)	8 (61.54)	
Unilateral Achilles clonus	6 (6.12)	3 (17.65)	3 (20.00)	2 (15.38)	<0.001
Bilateral Achilles clonus	9 (9.18)	8 (47.06)	5 (33.33)	5 (38.46)	
Penn UMN score, mean (SD)	6.83 (5.5)	16.7 (5.76)	16.33 (7.70)	12.57 (6.13)	<0.001
Fasciculations, single	19 (19.39)	5 (29.41)	2 (13.33)	0 (0.00)	<0.001
Fasciculations, focal continuous	51 (52.04)	10 (58.82)	1 (6.67)	0 (0.00)	
Fasciculations, multifocal continuous	7 (7.14)	0 (0.00)	0 (0.00)	0 (0.00)	
Cramps (presence)	35 (35.71)	8 (47.06)	1 (6.67)	2 (15.38)	0.026
Ashworth scale score at sampling	0.58 ± 0.93	2.63 ± 0.62	2.67 ± 0.98	1.83 ± 0.82	<0.001
CS tract hyperintensity (MRI) (118 patients)	14 (17.28) ^1^	7(63.63) ^2^	7 (46.67) ^3^	3 (27.27) ^4^	0.002
Prolonged central motor conduction time (MEP) (114 patients)	56 (74.67) ^5^	13 (100.00) ^6^	12(85.71) ^7^	10(83.33) ^8^	0.178

Values are means ± SD or absolute numbers (%). ^1^ Among 98 ALS, 81 underwent brain MRI for detecting CS tract hyperintensity; ^2^ Among 17 UMNp-ALS, 11 underwent MRI; ^3^ All 15 PLS underwent MRI; ^4^ Among 13 hSP, 11 underwent MRI. ^5^ Among 98 ALS, 75 underwent motor evoked potentials (MEP); ^6^ Among 13 UMNp-ALS, 11 underwent MEP; ^7^ Among 15 PLS, 14 underwent MEP; ^8^ Among 13 HSP, 12 underwent MEP.

**Table 3 biomedicines-09-01623-t003:** pNfH in CSF and serum in different clinical groups characterized by UMN involvement.

pNfH	ALS (*n* = 95)	UMNp ALS (*n* = 20)	PLS (*n* = 15)	hSP (*n* = 13)
In CSF (ng/mL)	2.09 [1.43–3.42]	1.94 [1.62–3.65] ALS vs. UMNpALS: *p* = 0.827	1.20 [0.3–1.78] ALS vs. PLS: *p* < 0.001 UMNp vs PLS: *p* < 0.001	0.43 [0.22–0.71] ALS vs. hSP: *p* < 0.001 UMNp vs. hSP: *p* < 0.001 PLS vs. hSP: *p* < 0.030
In serum (pg/mL)	125.88 [43.89–283.63]	137.77 [42.9–313.46] ALS vs. UMNpALS: *p* = 0.746	79.78 [10–148.95] ALS vs. PLS: *p* = 0.07 UMNp vs. PLS: *p* = 0.06	2.06 [0.1–22.7] ALS vs. hSP: *p* < 0.001 UMNp vs. hSP: *p* < 0.001 PLS vs. hSP: *p* < 0.013

Values are expressed as medians [interquartile range].

**Table 4 biomedicines-09-01623-t004:** Receiver operating characteristic curves for serum and CSF pNfH for discriminating ALS from other UMN syndromes.

pNfH ^1^	ALS vs. PLS AUC (CI)	UMNp ALS vs. PLS AUC (CI)	ALS vs. hSP AUC (CI)	PLS vs. hSP AUC (CI)	UMNp ALS vs. hSP AUC (CI)
In CSF (ng/mL)	0.75 (0.61–0.88)	0.74 (0.61–0.89)	0.95 (0.90–0.99)	0.72 (0.52–0.93)	0.97 (0.91–1.00)
In serum (pg/mL)	0.66 (0.52–0.80)	0.75 (0.56–0.94)	0.86 (0.77–0.95)	0.79 (0.62–0.97)	0.93 (0.84–1.00)

CI = confidence intervals. ^1^ crt-pNfH concentrations.

**Table 5 biomedicines-09-01623-t005:** Prognostic role of clinical and biological features in tracheostomy-free survival in all-groups analysis.

Variable	HR	95% CI	*p* > |z|
Sex (male/female)	1.33	0.84–2.11	0.21
Diagnostic delay (months)	0.91	0.88–0.94	<0.01
Time from onset to sampling (months)	0.94	0.91–0.96	<0.01
Age at sampling (years)	1.00	0.99–1.02	0.39
Site of onset (bulbar, upper limb, lower limb, respiratory)	0.72	0.55–0.95	0.02
Time to generalization (months)	0.96	0.94–0.98	<0.01
BMI at sampling (kg/m^2^)	1.01	0.96–1.05	0.66
ALSFRS-R score at sampling (total score)	0.97	0.94–1.00	0.04
Ashworth score at sampling (total score)	0.65	0.52–0.80	<0.01
Progression rate at sampling (points/month)	2.26	1.83–2.78	<0.01
Clinical subgroups (hSP/PLS/UMNp ALS/ALS)	0.24	0.14–0.41	<0.01
Dementia	2.71	1.44–5.07	<0.01
Serum pNfH ^(1)^, pg/mL	1.56	1.30–1.86	<0.01
CSF pNfH ^(1)^, ng/mL	50.54	16.72–152.78	<0.01

Values are means ± SD. Univariate analysis. ^(1)^: crt-pNfH concentrations.

## Data Availability

The data that support the findings of this study are available from the corresponding author, E.Z., upon reasonable request.

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
