# Peer review of "CSF Heavy Neurofilament May Discriminate and Predict Motor Neuron Diseases with Upper Motor Neuron Involvement"

_biomedicines, 2021, doi:10.3390/biomedicines9111623_

Round 1
Reviewer 1 Report
The present study is going to understand the phosphorylated neurofilament heavy chain (pNfH) can discriminate different upper motor neuron (UMN) syndromes in 281 patients. I like to give the following comments.
- Abbreviation needs to show in clear. What is ROC analysis n abstract?
- The clinical progression is important in comparison because healthy controls did not include. In Table 1, progression rate was markedly different in the data.
- Table 3 needs the variations between groups. Please add the p value.
- Case number between UMNp ALS and ALS (95 vs 20) seems too marked.
- The diagnostic delay for PLS has been discussed as the factor contrasted with previous report [21]. How to prevent it?
- Merits and bias in the application of pNfH level must discuss in detail.
- Please show the conclusion and novelty in clear.
Reviewer 2 Report
The authors have measured pNfH levels in CSF and serum of patients presenting with UMN signs, who were later diagnosed within a definite clinical group after long follow up period. In authors previous paper (Zucchi E, Neurodegener Dis 2018;18:255-261), though this a pilot study, the authors have initially supported phosphorylated neurofilament heavy chain (pNfH) may discriminate different UMN syndromes at diagnosis. Then they subsequently completed this study we have now seen. To a large extent, this study had larger sample and provided a cutoff value of CSF pNfH to differentiate ALS, UMNp-ALS included from PLS and HSP.
Their study is well designed and does a great job of concisely explaining and addressing the hypothesis and results.
I welcome this paper but would recommend the following revisions prior to publication:
1. The title of this paper was insufficient to highlight the main conclusion and significance. If possible, please try to re-write the title ensuring that it is informative and enlightening.
2. The first main finding of this study as summarized in section of Discussion by authors, the first is the results further supported the CSF and serum concentrations of pNfH may aid in the differentiation of UMN syndromes with a more favourable prognosis, as reported in their previous study. Indeed, the concentrations of pNfH between PLS and ALS, UMNp-ALS included were statistically different, and the lower concentrations of NfL at diagnosis may be more likely to support one’s diagnosis was PLS. Interestingly, I have noticed that in Table 1, subjects’ clinical features among PLS group, ALS group, and UMNp-ALS group were very distinct, for example, the “onset to sampling” and “Diagnostic latency” two items, “onset to sampling” of PLS was 59.29 months, while that was 16.46 months for UMNp-ALS and 12.57 months for ALS. the most obvious reason was that the faster progression rate and weakness symptoms leads people much earlier and more likely to seek medical help; in clinical practice, comparing some ambiguous differences of the CSF pNfH concentrations between PLS and ALS, the diagnostic latency was more practical, notable and worth highlighting. Considering the Elisa test which was used in this study could vary considerably caused by different laboratory environment.
3. In section of Discussion, line 288, “This finding for CSF contrasts with what was published by Verde and colleagues where pNfH was not significantly different in PLS compared to ALS. Why the two study have inconsistent findings on this issue? Was it correlated with the different disease phases of chosen participants? Previous studies have found serum pNfH, as well as NfL, elevated well before the time of diagnosis in mainly sporadic ALS patients, then continued elevated depend on the disease progression in early stage of disease course (about 2 years), and might kept relatively stable levels in middle or late stage of disease course. Have the authors consider this issue could have effects on the results? I’d like to read it in the section of Discussion.
4. The second main results “our results may support the notion that PLS is a different nosological entity from the ALS disease spectrum, where UMN-p ALS had similar pNfH levels compared to the other phenotypes of ALS, flail forms included”. In my opinion, only this difference in pNfH levels were far from enough to support PLS to be regarded as a different nosological entity. In fact, this levels mainly influenced by the severity of axonal damage and the speed of axonal damage, reflecting a state of dynamic inequilibrium in which protein synthesis, protein release into blood, turnover, and immune reaction are unbalanced with degradation. Please re-write this sentence.
5. Instead, the conclusion “Its prognostic role is confirmed in classic and bulbar ALS, but not among UMNp, where clinical signs remained the only independent prognostic factors.” was more valuable. These results were consistent with that reported by Gaiani A, et al (Gaiani A, JAMA neurology. 2017;74(5):525-532). The results well explained “why CSF levels of NfL were significantly increased in patients with upper motor neuron-dominant (UMND) ALS, despite the widely shown almost best prognosis associated with this subtype”.
6. Additionally, I have not found the Table 4 in the PDF, unless I have neglected it (my apologies if so!), could you please provide it?
7. There were two spelling “HSP and “hSP” in the paper, was it necessary? If not, please unify the expression.
